

# Contribution of patchy reconnection to the ion to electron temperature ratio in the Earth's magnetotail

Chuxin Chen[1], Chih-Ping Wang[2]

[1]CAS Key Laboratory of Geospace Environment, Department of Geophysics and Planetary Sciences, School of Earth and
Space Sciences, University of Science and Technology of China, Hefei, Anhui, 230026, China
[2]Department of Atmospheric and Oceanic Sciences, University of California, Los Angeles, CA, USA

*Correspondence to*: Chuxin Chen (chuxin@ustc.edu.cn)

**Abstract.** The ion to electron temperature ratio is a good indicator of the processes involved in the plasma sheet. Observations have suggested that patchy reconnection and the resulting earthward bursty bulk flows (BBFs) transport may be involved in causing the lower temperature ratios at smaller radial distances during southward IMF periods. In this paper, we theoretically estimate how patchy magnetic reconnection electric field accelerates ions and electrons differently. If both ions and electrons are non-adiabatically accelerated merely once in a single reconnection, the temperature ratio would be preserved. However, the ratio would not be preserved if particles are accelerated multiple times. As particles are transported earthward by BBFs after reconnection, the reflection of electrons from the ionosphere and subsequently multiple non-adiabatic accelerations at the reconnection site can explain the observed lower temperature ratios closer to the Earth.

## 1 Introduction

Among the parameters that are sensitive to the physical processes taking place in the plasma sheet, the ion to electron temperature ratio ($T_i/T_e$) is one of the most important, and the variation of this ratio is affected by the mechanisms involved. Adiabatic processes would preserve the $T_i/T_e$ ratio. How non-adiabatic processes would change the $T_i/T_e$ ratio remains to be investigated.

Using THEMIS data, Wang et al. (2012) analyzed the dependence of the ion to electron temperature ratio on the local time and radial distance from the Earth, and found that the ratio $T_i/T_e$ decreases from ~4 to ~2 as AE increases from 10 to 1000. In addition, when plasma sheet plasma is warm during southward IMF period, a comparison of the specific entropies between the distant- and near-Earth tail suggested that non-adiabatic processes energize the electrons more than the ions, which leads to a lower $T_i/T_e$ ratio at smaller radial distances. Earthward bursty bulk flows (BBFs), which can be generated by patchy reconnection (i.e., short time duration and short X-line length) in the plasma sheet, were observed to occur more frequently when AE is higher (e.g., Wang et al., 2009). How ions and electrons are accelerated differently by the reconnection electric field is not fully understood. Patchy reconnection also results in plasma of reduced total entropy (plasma bubble) than its surrounding. This entropy difference moves the bubble plasma earthward as BBFs (Chen and Wolf,





1999) and can thus change the temperature ratio at smaller radial distances. It is therefore the intent of the present study to give a theoretical estimation on how the $T_i/T_e$ ratio would be changed by patchy magnetic reconnection electric field and how the resulting BBF transport alter the radial profile of the ratio from midtail to the near-Earth.

This paper is organized as follows. In Section 2, a set of equations describing the non-adiabatic acceleration of
charged particle by magnetic reconnection electric field is derived. In Section 3, we calculate the $T_i/T_e$ ratios in the plasma sheet as a function of radial distances from X = –60 to –10 $R_E$ using the derived equations and empirical magnetospheric plasma and magnetic field models. We compare the computed ratios with the observed ratios to evaluate the importance of the contribution from patchy reconnection. Finally, discussion and summary are provided in Section 4.

**2 Non-adiabatic Acceleration of Charged Particles by Patchy Magnetic Reconnection Electric Field**

Patchy magnetic reconnection may take place on the dayside magnetopause when the IMF is southward, since the typical duration of magnetic flux transfer events (FTEs) was found to be approximately 1 to 2 min (Rijnbeek et al., 1984). When the IMF is northward, the duration of high-latitude reconnection is on the order of a couple of minutes (Onsager et al., 2001). Patchy magnetic reconnection may also take place at the nightside far tail as the ionospheric signature of magnetic reconnection, such as the auroral poleward boundary intensifications (PBIs), is short lived at 1–3 min (Zesta et al., 2002).
Patchy reconnections may become frequent during substorms (Chen, 2016). In this section we investigate theoretically the changes in the ion to electron temperature ratio at the tail X-line due to the acceleration by the patchy magnetic reconnection. Here we consider only the most essential reconnection element, that is, reconnection electric field, and do not include other secondary elements, such as waves.

Figure 1 shows the left part of an X-line in the magnetotail. In the following explanation, we will focus in more
detail on the upper half of this picture. In our calculations, the reconnection electric field is represented by $E$ , the x-extension of the cross section of the flux tube is $\Delta x$ , the X-line length is $\Delta y$ (Chen, 2013; Nakamura et al., 2004), the time duration of reconnection is $\Delta t$ , the number density is $n$ , the magnetic field is $B$ , and the ion and electron temperatures in the pre-reconnected flux tube are $T_i$ and $T_e$ , respectively. If we assume that the z-extension of the domain of the non-adiabatic motion of the ions is (i.e., on the order of ion gyroradius in the magnetic field $B$ due to the meandering motion
around the neutral sheet in the yz-plane (Hill, 1975; Speiser, 1965)):

$$a_i = \frac{m_i \overline{U_i}}{eB} ,$$ (1)

where $m_i$ is the mass of the proton, $\overline{U_i}$ is the mean thermal speed of the proton, $e$ is the electric charge of the proton. Assuming the z-extension of the domain of the non-adiabatic motion of the electrons (i.e., on the order of electron gyroradius) is:


$$a_e = \frac{m_e \overline{U_e}}{eB}$$ (2)



where $m_e$ is the mass of electron, and $\overline{U_e}$ is the mean thermal speed of electrons. Then total number of ions passing through the domain of the non-adiabatic motion of the ions should be:

$$\Delta N_i = n\Delta x \Delta y a_i \frac{\Delta t}{\tau_i},$$ 
(3)

where:

$$\tau_i = \frac{\pi r_i}{\overline{U_i}}$$ 
(4)

is the time an ion will stay in the domain of the non-adiabatic motion of the ions, or the half ion cyclotron (in the xy-plane) period, and:

$$r_i = \frac{m_i \overline{U_i}}{eB_z}$$ 
(5)

is the ion gyroradius in the finite z-directional magnetic field $B_z$. Substituting Eqs. (1), (4), and (5) into (3), we obtain:

$$\Delta N_i = n\Delta x \Delta y \Delta t \frac{\overline{U_i}}{\pi} \frac{B_z}{B}.$$ 
(6)

Likewise, the total number of electrons passing through the domain of the non-adiabatic motion of the electrons should be:

$$\Delta N_e = n\Delta x \Delta y a_e \frac{\Delta t}{\tau_e},$$ 
(7)

where:

$$\tau_e = \frac{\pi r_e}{\overline{U_e}}$$ 
(8)

is the time an electron will stay in the domain of the non-adiabatic motion of the electron or the half electron cyclotron (in the xy-plane) period, and:

$$r_e = \frac{m_e \overline{U_e}}{eB_z}$$ 
(9)

is the electron gyroradius in the finite z-directional magnetic field $B_z$. Substituting Eqs. (2), (8), and (9) into (7), we obtain:

$$\Delta N_e = n\Delta x \Delta y \Delta t \frac{\overline{U_e}}{\pi} \frac{B_z}{B}.$$ 
(10)

Although the particles have an equal chance to pass through the upper or lower portions of flux tube, for simplicity, we assume that the particles move back to their original location (the same argument applies to the lower part of the X-line). Each ion will have an energy increment through the domain of non-adiabatic motion of:

$$\Delta w_i = 2r_i Ee,$$ 
(11)



and each electron will have an energy increment through the domain of non-adiabatic motion of:

$$\Delta w_e = 2r_e E e . \tag{12}$$

The total energy increment of the ions through the domain of non-adiabatic motion will be:

$$\Delta W_i = \Delta N_i \Delta w_i . \tag{13}$$

Substituting Eqs. (5), (6), and (11) into (13), we obtain:

$$\Delta W_i = 2n\Delta x \Delta y \Delta t E \frac{m_i \overline{U_i}^2}{\pi B} , \tag{14}$$

where:

$$\overline{U_i} = \sqrt{\frac{8kT_i}{\pi m_i}} \tag{15}$$

and $k$ is Boltzmann's constant. Substituting Eq. (15) into (14), we have

$$\Delta W_i = 16n\Delta x \Delta y \Delta t E \frac{kT_i}{\pi^2 B} . \tag{16}$$

The total energy increment of the electrons through the domain of non-adiabatic motion would be:

$$\Delta W_e = \Delta N_e \Delta w_e . \tag{17}$$

Substituting Eqs. (9), (10), and (12) into (17), we obtain:

$$\Delta W_e = 2n\Delta x \Delta y \Delta t E \frac{m_e \overline{U_e}^2}{\pi B} , \tag{18}$$

where:

$$\overline{U_e} = \sqrt{\frac{8kT_e}{\pi m_e}} . \tag{19}$$

Substituting Eq. (19) into (18), we obtain:

$$\Delta W_e = 16n\Delta x \Delta y \Delta t E \frac{kT_e}{\pi^2 B} . \tag{20}$$

Let the total number of ions or electrons be $N$ in the upper part of the flux tube. Then, the total energy of the ions

before reconnection is:

$$W_i = \frac{3}{2} NkT_i , \tag{21}$$

and the total energy of the electrons before reconnection is:

$$W_e = \frac{3}{2} NkT_e . \tag{22}$$

The ion to electron temperature ratio in the flux tube after reconnection will then be:



$$\frac{T_{i,a}}{T_{e,a}} = \frac{W_i + \Delta W_i}{W_e + \Delta W_e} = \frac{\frac{3}{2}NkT_i + 16n\Delta x\Delta y\Delta tE\frac{kT_i}{\pi^2 B}}{\frac{3}{2}NkT_e + 16n\Delta x\Delta y\Delta tE\frac{kT_e}{\pi^2 B}} = \frac{T_i}{T_e},$$ (23)

where the subscript $a$ represents the time after reconnection.

Equation (23) indicates that the ratio $T_i/T_e$ is preserved after a patchy magnetic reconnection. However, this is only true when the temperature of both the ions and electrons in Eqs. (16) and (20) are constant during the patchy reconnection. This condition does not hold once the accelerated particles return to the same domain of non-adiabatic motion after reflection at the mirror point.

In that case, Eq. (16) becomes:

$$\Delta W_i = 16n\Delta x\Delta yE\frac{k}{\pi^2 B}\left(T_i\tau_i + T_it_{i,roundtrip,1} + T_{i,1}\tau_i + T_{i,1}t_{i,roundtrip,2} + \cdots\right),$$ (24)

where:

$$t_{i,roundtrip,l} = 2\frac{L}{\overline{U_{i,l}}}$$ (25)

is the round trip traveling time of the ions in the distance $L$ from the reconnection site to the mirror point and:

$$\overline{U_{i,l}} = \sqrt{\frac{8kT_{i,l}}{\pi m_i}},$$ (26)

where:

$$T_{i,l} = T_{i,(l-1)} + \frac{4r_{i,(l-1)}Ee}{3k},$$ (27)

and the subscript $l$ represents the $l$ th time non-adiabatic acceleration.

Likewise, Eq. (20) becomes:

$$\Delta W_e = 16n\Delta x\Delta yE\frac{k}{\pi^2 B}\left(T_e\tau_e + T_et_{e,roundtrip,1} + T_{e,1}\tau_e + T_{e,1}t_{e,roundtrip,2} + \cdots\right),$$ (28)

where:

$$t_{e,roundtrip,l} = 2\frac{L}{\overline{U_{e,l}}}$$ (29)

is the round trip time of the electron and:

$$\overline{U_{e,l}} = \sqrt{\frac{8kT_{e,l}}{\pi m_e}},$$ (30)

where:



$$T_{e,l} = T_{e,(l-1)} + \frac{4 r_{e,(l-1)} E e}{3k} . \tag{31}$$

Then, the ion to electron temperature ratio will be:

$$\frac{T_{i,a}}{T_{e,a}} = \frac{W_i + \Delta W_i}{W_e + \Delta W_e} \neq \frac{T_i}{T_e} . \tag{32}$$

It is noteworthy that because the number density $n$ , the x-extension of the cross section of the flux tube $\Delta x$ , the X-line length $\Delta y$ , appear both explicitly or implicitly in the numerator and denominator of Eqs. (23) and (32), the ion to electron temperature ratio calculated by these two equations are independent of the quantities $n$ , $\Delta x$ and $\Delta y$ .

## 3 Variation of Temperature Ratio From Midtail to Near-Earth Tail

To quantitatively estimate how the acceleration by patchy tail reconnection electric field and the resulting earthward BBF transport alter the radial profile of the ion to electron temperature ratio in the plasma sheet, we compute Eq. (32) by using two empirical magnetospheric models. The Tsyganenko T96 magnetospheric magnetic field model (Tsyganenko, 1995) and Tsyganenko tail plasma sheet model (Tsyganenko and Mukai, 2003) are used for magnetic field and plasma, respectively, where the input parameters are: a solar wind density of 6 cm$^{-3}$, velocity of 380 km/s, IMF $B_z$ = -5 nT, and $Dst$ = -80 nT.

Figure 2 shows schematically the procedure of our calculation. From the T96 magnetic field model, an equatorial point $x_1$ is first located. Then the magnetic field line passing through this point is traced out to the ionosphere, and the length $2L$ of this field line is recorded. The plasma data on this field line is determined from the Tsyganenko tail plasma sheet model.

Assuming a patchy magnetic reconnection occurs at the mid-point $x_{half}$ of this field line (Borovsky et al., 1998), the cut flux tube moves earthward as bursty bulk flows. The total entropy of the earthward half of the cut flux tube is calculated and compared with the entropies of the equatorial plane points with the same y value. The earthward motion would stop at a location where the total entropy of background plasma is the same as the entropy of the cut flux tube. The x value of the matched point would be the final position $x_{final}$ of the post-reconnected flux tube (Chen and Wolf, 1999). In our calculation, we bear in mind that BBFs can be responsible for > 60% of earthward transport of mass and energy when AE is high (Angelopoulos et al., 1994), and BBFs are produced by magnetic reconnection. In this simplified picture, for plasma to be transported from the far tail to the near-Earth tail, magnetic reconnection on the same flux tube is required. For example, a flux tube extending to X = –48 and Y = 0 is cut by a reconnection at X = –24 and Y = 0 and will then move to its final position at X = –10.2 and Y = 0. Only when the total entropy of a flux tube is reduced, it is possible for a cut flux tube to move earthward. The friction or turbulence heating effect of BBF is not considered in the present study.

Figure 3 shows one of our calculations. In this run, for the reconnection the electric field $E$ was set to 4 mV/m, the time duration of reconnection $\Delta t$ was set to 60 s, and the z-directional magnetic field $B_z$ was set to 0.5 nT. As can be seen





from the equations in Section 2, these quantities are not independent of each other. We also experimented with other values of $E$ from 4 mV/m to 10 mV/m, $\Delta t$ from 40 s to 80 s, and $B_z$ from 0.5 nT to 1.0 nT, and the results are similar.

The domain of our calculation (x = –10 to –60 $R_E$ and y = –20 to 20 $R_E$ ) is limited by the domain of the Tsyganenko tail plasma sheet model, which is valid for 10~50 $R_E$ down the Earth tail. We linearly interpolated the equatorial

plasma data between 50 $R_E$ (the tailward boundary of Tsyganenko tail plasma sheet model) and 220 $R_E$ down the tail with the observed plasma number density of 0.3 cm$^{-3}$ and an electron temperature of 1.2 x 10$^6$ $K$ at 220 $R_E$ (Slavin et al., 1985).

The magnetic field strength at the mid-point of each pre-reconnected field line is represented as $B$ in the calculation. The electron temperature was set to be six times smaller than the ion temperature (i.e., close to the ratio in the magnetosheath) in each pre-reconnected field line.

The calculated ion to electron temperature ratios in the equatorial plane shown in Figure 3 are comparable to observational values shown in the left (warm plasma) and right panels (cool and warm combined) of Figure 4 (the observed asymmetry between dawn and dusk is not considered in present theoretic work). In particular, the closer a position is to the Earth, the lower the ratio is. The ratio is slightly higher toward dawn or dusk than at midnight. It is noteworthy that the $T_i/T_e$ ratio is preserved tailward of 60 $R_E$ because of the large distance between the reconnection site and the ionosphere.

This comparison thus suggests that patchy magnetic reconnection electric field and the resulting BBF transport are likely to be the main contributor to the observed radial variation in the temperature ratio during southward IMF periods.

**4 Discussion and summary**

Our study demonstrated that patchy tail reconnection electric field and the resulting BBF transport can be important to the radial variation of the ion to electron temperature ratio from the midtail to the near-Earth tail during southward IMF periods.

A significant difference between our present study and those of Onsager et al. (1991) and Schriver et al. (1998) is the electromagnetic field experienced by charged particles in the plasma sheet. In our study, the electric field was not assumed to be constant and steady, but was enhanced in some local channels (with varying length from the Earth) and short lived. The acceleration of charged particles in such a spatially and temporally short electric field is fundamentally different from that under an uniform and steady one. Besides the electric field, the magnetic field experienced by charged particles is

also much different. After leaving the reconnection site, a flux tube is dipolarized, in contrast to the steady one. The motion of charged particle in such dipolarized field lines is adiabatic. The reconnection sites are not necessarily confined only to the far tail and can be possibly everywhere in the plasma sheet.

The location of a reconnection site, or more precisely the distance from a reconnection site to the ionosphere, plays a crucial role in the preservation or variation of the ion to electron temperature ratio. The non-preservation of the $T_i/T_e$ ratio

comes from the terms higher than the second one on the righthand side of Eqs. (24) and (28). In other words, when the effective time duration of reconnection is less than the sum of the half cyclotron period and the round trip travel time from the reconnection site to the ionosphere for both the accelerated ions and electrons, the ratio is preserved. Since the much





lighter mass of electron compared to that of ion, the cyclotron period is much less for electron than it is for ion, as can be seen from Eqs. (4) and (8). For the observed $T_i/T_e$ ratio in the magnetosphere, electrons have a much higher thermal speed than ions. Consequently, the round trip travel time from the reconnection site to the ionosphere and back is much less for electron than it is for ion, as can be seen from Eqs. (25) and (29). In the region earthward of 60 $R_E$, ions experience non-adiabatic acceleration no more than once, while most electrons experience non-adiabatic acceleration more than once, which leads to a lower $T_i/T_e$ ratio than that of the magnetosheath.

The ionosphere could be an energy sink for the magnetospheric particles. For the study of the ion to electron temperature ratio, which requires fractional particles exchange between the magnetosphere and the ionosphere in short time (tens of minutes), this effect is negligible, because the number flux of precipitation particle is too small to accomplish this task. For example, taking the precipitation number flux as 7.93 x 10⁶ cm⁻²sec⁻¹sr⁻¹(Hardy et al., 1989), a flux tube extending to 20 $R_E$ down tail with a number density of 0.3 cm⁻³ will take about 140 hours to be replaced. Thus the ionosphere should be treated only as a reflector to the magnetospheric particles in present investigation.

The contours of the ion to electron temperature ratio on the equatorial plane are dawn-dusk symmetric since we did not include the energetic particle's gradient and curvature drifts in the calculation. It is noteworthy that particle's gradient and curvature drift may violate the conservation of total entropy as plasma moving earthward.

In summary, we demonstrated that patchy magnetic reconnection electric field may preserve the ion to electron temperature ratio if charged particles are non-adiabatically accelerated no more than once in a single reconnection, while multiple occurrences of non-adiabatic acceleration vary the ratio. The observed lower $T_i/T_e$ ratio near the Earth is therefore caused by the electrons reflected back from the ionosphere and undergoing multiple non-adiabatic accelerations.

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



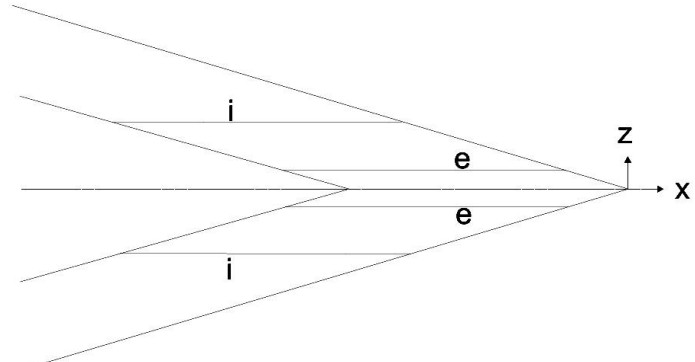

**Figure 1. Sketch showing the left part of an X-line. The oblique magnetic field lines are in the xz-plane, and the x-axis indicates the neutral sheet. The area between the line labeled "e" and the neutral sheet represents the non-adiabatic motion domain for electrons, and the area between the line labeled "i" and the neutral sheet represents the non-adiabatic motion domain for ions.**

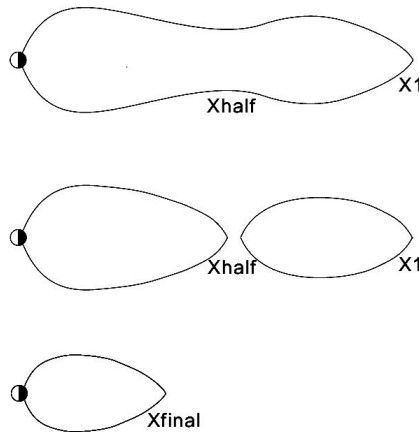

**Figure 2. Sketch showing patchy magnetic reconnection and the related positions. A flux tube whose $T_i/T_e$ ratio is under investigated, originally extending to $x_1$ ( $x_1 = 2\,x_{half}$ ), is cut by a reconnection taking place at $x_{half}$ , having its final position at $x_{final}$ .**





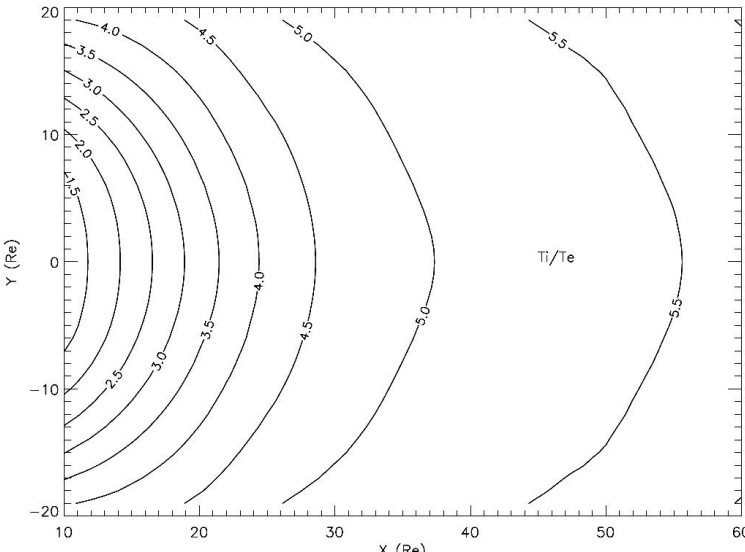

**Figure 3. Contours of the calculated ion to electron temperature ratio on the equatorial plane.**

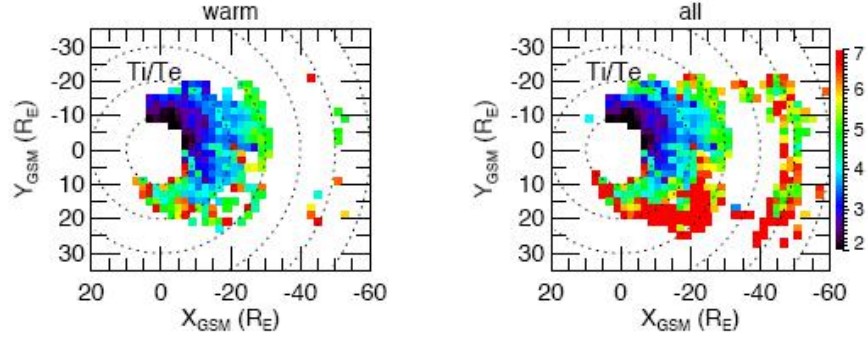

**Figure 4. The observed $T_i/T_e$ ratio in the plasma sheet on the X-Y plane. The left panel is for warm plasma, and the**

5 **right panel is for the combination of cool and warm plasma.**