# Peer review of "Contribution of patchy reconnection to the ion to electron temperature ratio in the Earth's magnetotail"

_Annales Geophysicae, 2018_

## Referee Comment (RC1) · Anonymous Referee #1 · 30 Jun 2018

The paper presents a study that calculates Ti/Te ratio. The authors conclude that Ti/Te ratio would be preserved if reconnection only happens once. This conclusion is not supported, as outlined below. The biggest problem with the study is the methodology. The authors model reconnection by simply cutting the Tsyganenko magnetic field by half, ignoring the heating associated with reconnection. The authors claim that Ti/Te would be preserved if the magnetic field line is cut by half. Actually, given the methodology and assumptions, Ti/Te should still be preserved even when the reconnection happens more than once to the same field line. Can the authors show that using the same methodology, if reconnection happened more than once, Ti/Te would not be preserved? Why would the authors introduce reconnection? Is this necessary? Why not

just try the methodology without reconnection? The authors claim that the comparison with the observations (Figure 4) is good, but upon close examination, there are some differences. The authors need to discuss these differences. Tsyganenko model is a time independent model with no reconnection built in. How do the authors model the field line after the reconnection in Figure 2? Tsyganenko magnetic field does not cross the equatorial plane at Xhalf in Figure 2. How do the authors force the model to bend the field lines so that they reconnect at Xhalf and how would the authors know what the field line configuration would look like after the reconnection?

It is not clear if the formalism presented by the authors can handle reconnection.

---

## Author Comment (AC1) · 2 Jul 2018

Reply to Referee #1:

Referee #1's comment:

The paper presents a study that calculates Ti/Te ratio. The authors conclude that Ti/Te ratio would be preserved if reconnection only happens once. This conclusion is not supported, as outlined below. The biggest problem with the study is the methodology. The authors model reconnection by simply cutting the Tsyganenko magnetic field by half, ignoring the heating associated with reconnection. The authors claim that Ti/Te

would be preserved if the magnetic field line is cut by half. Actually, given the methodology and assumptions, Ti/Te should still be preserved even when the reconnection happens more than once to the same field line. Can the authors show that using the same methodology, if reconnection happened more than once, Ti/Te would not be preserved? Why would the authors introduce reconnection? Is this necessary? Why not just try the methodology without reconnection?

Our response:

We are very grateful to Referee #1 for helpful comments.

There is a misleading in this matter. We did not state reconnection only happens once. Reconnection can happen more than once in our scenario in order to transport reconnected flux tube from far tail to the near-Earth tail by bursty bulk flows (BBFs). Eq. (23) shows Ti/Te is conserved after reconnection. But if considering the additional non-adiabatic acceleration of particle associated with the round trip from the mirror point (see Eq. (24)), then Ti/Te will not be conserved when the round trip happens more than once in a single reconnection. The heating associated with reconnection is considered in our calculation as Eqs. (11) and (12). It is the difference of the heating between ions and electrons at each reconnection site, which leads to the lower Ti/Te ratio close to the Earth. We will restate the "the non-adiabatic acceleration of particle happens no more than once in a single reconnection" as "the non-adiabatic acceleration of particle happens no more than once in each reconnection" in the revised version.

As we have described in Introduction, the reason we consider reconnection as one of the possible processes is based on the understanding established from previous studies: (1) the observed Ti/Te is correlated with AE levels, (2) AE levels are correlated with occurrence of BBFs, and (3) patchy reconnection can generate BBFs. If only consider the background slow earthward convection without reconnection generated BBFs, then it only results in adiabatic energization and Ti/Te ratio is conserved.

Referee #1's comment:

The authors claim that the comparison with the observations (Figure 4) is good, but upon close examination, there are some differences. The authors need to discuss these differences. Tsyganenko model is a time independent model with no reconnection built in. How do the authors model the field line after the reconnection in Figure 2? Tsyganenko magnetic field does not cross the equatorial plane at Xhalf in Figure 2. How do the authors force the model to bend the field lines so that they reconnect at Xhalf and how would the authors know what the field line configuration would look like after the reconnection?

It is not clear if the formalism presented by the authors can handle reconnection.

Our response:

We understand the limitation for comparing theoretical calculation with observations, thus we did not claim that the comparison as "good". We have also discussed the differences in the comparison, such as the dawn-dusk asymmetry shown in the observation. Since we did not include the particle's gradient and curvature drift in the calculation, there is no asymmetry of Ti/Te between the dawn and dusk as indicated by the observations. Asides from this difference, there is a difference of the exact values of Ti/Te between our theoretical results and the observations. This difference can be reduced by choosing more suitable parameters in our calculation.

Since the observations of Ti/Te are statistical results, to theoretically estimate the Ti/Te and compare with the observations, we need to use a statistical magnetic field model. Our theoretical estimate of Ti/Te has only statistical meaning. The equatorial magnetic field after reconnection is assumed more dipolar (associated with BBFs and DF) than the background field lines. As stated in the discussion part, the motion of particles in this dipolarized field is adiabatic. For the two large-scale magnetic field quantities, the field-line length and the flux tube volume, used in our theoretical calculation, their values are contributed most by the distance from the equatorial location to its ionosphere. Thus, even though the local magnetic field changes near the equator after reconnection should affect the two quantities, the effects are not expected to be substantial. Therefore, despite that Tsyganenko magnetic field is time independent and without reconnection built in, it can empirically provide us reasonable estimates of these two quantities for both the magnetic fields associated with a reconnected flux tube and with the background plasma sheet.

Our formalism is based on the classical picture of reconnection of Hill 1975 and Speiser 1965. They are first-order estimate.
* * *

---

## Referee Comment (RC2) · Anonymous Referee #2 · 6 Jul 2018

The manuscript deals with ion to electron temperature ratio in the Earth's magnetotail. The authors present an analytical model to investigate the role of patchy reconnection from the near-Earth tail to mid-tail.

Specific comments:

1) Heating in reconnection I have significant concerns about how the authors treat (or actually seem to neglect) plasma heating in reconnection. The authors state on p5, lines 3-4: "Equation (23) indicates that the ratio Ti/Te is preserved after a patchy magnetic reconnection. However, this is only true when the temperature of both the ions and electrons in Eqs. (16) and (20) are constant during the patchy reconnection."

[Figure]

Plasma temperature does not stay constant during reconnection, patchy or extended.

Heating (i.e., increase in temperature) during reconnection has been the focus of several previous studies. For example, Drake et al. (JGR 2009; see also references therein) considered how ions get heated when they enter a reconnection exhaust (BBF). This has been further investigated by, e.g., Haggerty et al. (GRL 2015). I don't think this aspect of reconnection can be ignored when considering the ion to electron temperature ratio.

2) Discussion on the limitations of the model The present discussion is not comprehensive enough. Topics that should be discussed in much more detail than in the present manuscript include: a) dawn-dusk asymmetry The discussion on asymmetry should include both the exclusion of particle drifts in the model (already mentioned, but more detail needed), as well as the observational fact of "dusk side is more active" (more reconnection and reconnection-related phenomena are observed to take place on the duskside than on the dawnside, but this difference of occurrence rates is also not included in the model; (Raj et al., JGR 2002; Frey et al., JGR 2004; McPherron et al., JGR 2011; Nagai et al., JGR 2013; Liu et al., JGR 2013; Genestreti et al., JASTP 2014; Gabrielse et al., JGR 2014; Kiehas et al., JGR 2018)). b) scattering Waves are not included in the model, but what would be scattering's likely effects?

Technical corrections:

Figure 1: Does the x-axis in Figure 1 point towards the Earth or away from the Earth? In other words, is the flux tube shown in the figure Earthward or tailward of the X-line?

Figure 3: The figure requires several improvements so that comparisons could be made with the observations in Figure 4. The axis limits should match those of Fig. 4, and matching colors added. Furthermore, the labels should be bigger and overall resolution higher.

page 7, line 25: much -> very page 8, line 15: moving -> moves

---

## Author Comment (AC2) · 10 Jul 2018

Reply to Referee #2:

Referee #2's comment:

The manuscript deals with ion to electron temperature ratio in the Earth's magnetotail. The authors present an analytical model to investigate the role of patchy reconnection from the near-Earth tail to mid-tail.

Specific comments:

1) Heating in reconnection I have significant concerns about how the authors treat

(or actually seem to neglect) plasma heating in reconnection. The authors state on p5, lines 3-4: "Equation (23) indicates that the ratio Ti/Te is preserved after a patchy magnetic reconnection. However, this is only true when the temperature of both the ions and electrons in Eqs. (16) and (20) are constant during the patchy reconnection." Plasma temperature does not stay constant during reconnection, patchy or extended. Heating (i.e., increase in temperature) during reconnection has been the focus of several previous studies. For example, Drake et al. (JGR 2009; see also references therein) considered how ions get heated when they enter a reconnection exhaust (BBF). This has been further investigated by, e.g., Haggerty et al. (GRL 2015). I don't think this aspect of reconnection can be ignored when considering the ion to electron temperature ratio.

Our response:

We are very grateful to Referee #2 for helpful comments.

There is a misleading in this matter. The statement "when the temperature of both the ions and electrons in Eqs. (16) and (20) are constant during the patchy reconnection" really describes that ions and electrons sequentially entering the reconnection site during a patchy reconnection are of same temperature. Considering Eq. (24), when ions and electrons reentering the reconnection site are those accelerated in previous passing through the reconnection site and reflected back from the ionosphere, the ratio Ti/Te would not be preserved. The heating associated with reconnection is considered in our calculation as Eqs. (11) and (12). It is the difference of the heating between ions and electrons at each reconnection site leads to the lower Ti/Te ratio close to the Earth.

We will delete the statement "However, this is only true when the temperature of both the ions and electrons in Eqs. (16) and (20) are constant during the patchy reconnection" in a revised version to avoid confusion.

Referee #2's comment:

2) Discussion on the limitations of the model The present discussion is not comprehensive enough. Topics that should be discussed in much more detail than in the present manuscript include: a) dawn-dusk asymmetry The discussion on asymmetry should include both the exclusion of particle drifts in the model (already mentioned, but more detail needed), as well as the observational fact of "dusk side is more active" (more reconnection and reconnection-related phenomena are observed to take place on the duskside than on the dawnside, but this difference of occurrence rates is also not included in the model; (Raj et al., JGR 2002; Frey et al., JGR 2004; McPherron et al., JGR 2011; Nagai et al., JGR 2013; Liu et al., JGR 2013; Genestreti et al., JASTP 2014; Gabrielse et al., JGR 2014; Kiehas et al., JGR 2018)).

Our response:

We will include a more detail discussion about dawn-dusk asymmetry in a revised version as following. The finite cross-tail width of the plasma sheet and particle's gradient and curvature drift cause depletions of energetic ions on the dawn magnetopause and electrons on the dusk magnetopause, which is likely the main reason for the observed asymmetry of Ti/Te ratios with higher Ti/Te ratio on the duskside. Observations have suggested a dawn-dusk asymmetry with reconnection occurring more often on the dusk side (e.g., Raj et al., 2002; Kiehas et al., 2018). This asymmetry is not considered in our current model. Since our model shows a reconnection would lead to a lower Ti/Te ratio, such dawn-dusk asymmetric reconnection occurrence frequency is expected to reduce the dawn-dusk asymmetry caused by particle's gradient and curvature drift.

Referee #2's comment:

b) scattering Waves are not included in the model, but what would be scattering's likely effects?

Our response:

Wave scattering would mix the heated particles with those without in the same flux tube. By writing Eqs. (23) and (32), we implicitly include such effect of wave scattering in our model.

Referee #2's comment:

Technical corrections:

Figure 1: Does the x-axis in Figure 1 point towards the Earth or away from the Earth? In other words, is the flux tube shown in the figure Earthward or tailward of the X-line?

Our response:

The x-axis in Figure 1 points away from the Earth. The flux tube shown in the figure is earthward of the X-line.

Referee #2's comment:

Figure 3: The figure requires several improvements so that comparisons could be made with the observations in Figure 4. The axis limits should match those of Fig. 4, and matching colors added. Furthermore, the labels should be bigger and overall resolution higher.

Our response:

We will improve Figure 3 in a revised version.

Referee #2's comment:

page 7, line 25: much -> very page 8, line 15: moving -> moves

Our response:

We will make corrections in a revised version.